# Long-Term Supplementation of Ozonated Sunflower Oil Improves Dyslipidemia and Hepatic Inflammation in Hyperlipidemic Zebrafish: Suppression of Oxidative Stress and Inflammation against Carboxymethyllysine Toxicity

**DOI:** 10.3390/antiox12061240

**Published:** 2023-06-08

**Authors:** Kyung-Hyun Cho, Ji-Eun Kim, Ashutosh Bahuguna, Dae-Jin Kang

**Affiliations:** 1Raydel Research Institute, Medical Innovation Complex, Daegu 41061, Republic of Korea; 2LipoLab, Yeungnam University, Gyeongsan 38541, Republic of Korea

**Keywords:** ozonated sunflower oil (OSO), antioxidant, dyslipidemia, high-cholesterol diet (HCD), zebrafish, zebrafish embryo, carboxymethyllysine, CML

## Abstract

Ozonated sunflower oil (OSO) is a well-known functional oil with antioxidant, antimicrobial, anti-allergic, and skin-moisturizing properties. However, studies on the effects of OSO on high-cholesterol diet (HCD)-induced metabolic disorders have been scarce. In the current study, we aimed to determine the anti-inflammatory effects of OSO on lipid metabolism in adult hypercholesterolemic zebrafish and its embryos. Microinjection of OSO (final 2%, 10 nL) into zebrafish embryos under the presence of carboxymethyllysine (CML, 500 ng) protected acute embryo death up to 61% survival, while sunflower oil (final 2%) showed much less protection at around 42% survival. The microinjection of OSO was more effective than SO to inhibit reactive oxygen species (ROS) production and apoptosis in the CML induced embryo toxicity. Intraperitoneal injection of OSO under the presence of CML protected acute death from CML-induced neurotoxicity with improved hepatic inflammation, less detection of ROS and interleukin (IL)-6, and lowering blood total cholesterol (TC) and triglyceride (TG), while the SO-injected group did not protect the CML-toxicity. Long-term supplementation of OSO (final 20%, *wt*/*wt*) with HCD for 6 months resulted in higher survivability than the HCD alone group or HCD + SO group (final 20%, *wt*/*wt*) with significant lowering of plasma TC and TG levels. The HCD + OSO group showed the least hepatic inflammation, fatty liver change, ROS, and IL-6 production. In conclusion, short-term treatment of OSO by injection exhibited potent anti-inflammatory activity against acute neurotoxicity of CML in zebrafish and their embryo. Long-term supplementation of OSO in the diet also revealed the highest survivability and blood lipid-lowering effect through potent antioxidant and anti-inflammatory activity.

## 1. Introduction

Hyperinflammation is associated with many pathological processes of critical diseases, such as dyslipidemia [1] and autoimmune disease [2], and also initiated by proinflammatory signaling during infection or cell damage with non-specific attack [3,4], Dyslipidemia is characterized by elevated plasma levels of total cholesterol (TC), low-density lipoprotein cholesterol (LDL-C), or triglycerides (TG) or decreased plasma levels of high-density lipoprotein cholesterol (HDL-C) [5]. Consumption of a high-cholesterol diet (HCD) contributes to the progression of dyslipidemia, hepatic inflammation, and hepatic steatosis via elevation of serum TG and very low-density lipoproteins (VLDL)/LDL [6]. Dyslipidemia plays an important role to “add fuel to the fire” in the exacerbation of inflammatory cascade series, insulin resistance, type 2 diabetes, fatty liver change, and hepatic steatosis [7]. For decades, several studies have been conducted to evaluate the efficacy to treat simultaneously inflammation and dyslipidemia, because oxidative stress and proinflammatory signaling were more aggravated in hyperlipidemia, especially in high TG [8,9]. The critical role of dyslipidemia, oxidative stress, and inflammation has been observed in several diseases [10].

Ozone (O_3_), a bluish gas with three oxygen atoms that are easily broken into oxygen and oxygen atoms with high oxidizing power. Ozone has exhibited various medical effects, including antimicrobial, immunomodulatory, anti-tumor effect, and antioxidant effects, as well as epigenetic modification [11,12]. In particular, ozone has a strong germicidal effect with destructively against bacteria, fungi, and viruses: via ozonolysis of dual bonds of the cytoplasmic membrane and modify intracellular contents through protein oxidation, leading to the sterility [13]. Of note, at therapeutic doses, ozone has been demonstrated to inhibit the nuclear factor kappa B (NF-kB) efficiently. This event further influences inflammation by halting the activities of tumor necrosis factor-alpha (TNFα), interferon gamma (IFN γ), interleukin 6 (IL-6), and IL-8 [14]. Additionally, several ex vivo and in vivo studies have shown that ozone can activate the nuclear factor erythroid 2-related factor 2 (Nrf2) [15,16], which is responsible for the generation of antioxidant enzymes such as superoxide dismutase (SOD), haem-oxygenase 1(HO-1), and catalase. Recently ozone therapy was demonstrated to activate the casein kinase 2 (CK2)—a regulator of Nrf2, thereby protecting against oxidative stress in multiple sclerosis patients [17]. In another clinical study on multiple sclerosis patients, ozone was reported to modulate the Nrf2/NF-kB activities, strongly suggesting the antioxidant nature of ozone [17]. Conversely, ozone therapies displayed a great credential in reducing TC and LDL-C in cardiopathy [18], hypertension, and ischemic disease patients [19]. Another study showed that ozone treatment enhanced HDL-C decreased TC and TG, and inhibited inflammation in psoriatic patients [20]. Despite several medicinal benefits of ozone, its short life span is a matter of concern, which can be slightly improved by fixing the ozone in water. However, ozonated water cannot be stored for a long time because ozone in water is unstable with a short half-life [21].

To overcome the instability of ozone in water, ozonated oil was produced by the direct reaction of ozone with various vegetable oils, such as sunflower oil, almond oil, and olive oil, which are enriched in unsaturated fatty acids. Many ozonated oils in various vegetable oils claimed adequate wound healing activities or tissue regeneration in the skin via anti-microbial activity [22]. In a topical application, the ozonated oil acts as a moisturizer and protectant in patients with impaired skin barrier function [9].

Our previous study demonstrated that ozonated sunflower oil (OSO) has potent in vitro and cellular antioxidant properties that lead to a protective effect against hydrogen-peroxide-induced stress at the cellular level [23]. In an extension of the previous study, herein, we have performed an animal study to decipher OSO’s antioxidant and antiapoptotic properties against carboxymethyllysine (CML)-induced oxidative stress in zebrafish embryos. A further role of OSO was examined as an anti-inflammatory and preventive agent to improve dyslipidemia against the adversity posed by CML in high-cholesterol diet-supplemented adult zebrafish. Finally, the anti-obesity and lipid profile balancing effect of OSO against long-term supplementation of a high cholesterol diet was examined in zebrafish.

## 2. Materials and Methods

### 2.1. Materials

Ozonated sunflower oil (Raydel^®^ Bodyone Flambo oil) was provided by Rainbow and Nature Pty, Ltd. (Thornleigh, NSW, Australia). The characteristics of the OSO revealed the typical range of Oleozon^®^, as previously described [24]. Sunflower oil (SO) was purchased from the local supermarket in Daegu, South Korea. *N*-ε-carboxylmethyllysine (CAS-No 941689-36-7, Cat#14580-5g), dihydroethidium (DHE, 104821-25-2, Cat #37291), and acridine orange (AO, 65-61-2, Cat#A9231), oil red O (Cat#O0625), and 2-phenoxyethanol (Sigma P1126; St. Louis, MO, USA) were procured from Sigma-Aldrich (St. Louis, MO, USA). All other chemicals and reagents unless otherwise stated were of analytical grade and used as supplied.

### 2.2. Zebrafish Embryos Production

Zebrafish were maintained using the standard protocols as per the guidelines for the care and use of laboratory animals [25,26]. Zebrafish embryos were produced and collected as the previously described method [27]. In brief, female and male zebrafish were segregated from each other in the breeding tank using a physical divider. After 16 h of segregation, the divider from the breeding tank was removed to allow undisturbed mating for approximately 30 min. Embryos were collected, washed with egg water, and inspected for further experiments.

### 2.3. Microinjection of Zebrafish Embryos

Embryos at the 16-cell stage (1.5 h post-fertilization (hpf)) were randomly divided into six groups (each group n = 150–160) that received different treatments. Embryos in group I were injected with PBS (vehicle), while group II embryos were injected with 500 ng CML in PBS. Embryos in group III and group IV were injected with 500 ng CML along with the co-injection of SO1% and SO2%, respectively; similarly, embryos in group V and VI were injected with 500 ng CML with co-injection of OSO1% and OSO2%, respectively. The 1% and 2% doses were chosen because the selected concentration showed no adverse effect on embryos (mortality, and developmental deformities) while injected intraperitoneally (data not shown). The embryos in different groups received a constant treatment volume (10 nL), which was injected by microcapillary pipette using a pneumatic picopump (PV830; World Precision Instruments, Sarasota, FL, USA) equipped with a magnetic manipulator (MM33; Kantec, Bensenville, IL, USA). Bias was minimized by microinjections at the same position of the yolk. Embryos in all the groups were visualized under the stereomicroscope (Motic SMZ 168; Hong Kong) at 5 and 24 h post-treatment and images were captured by Motic cam2300 CCD camera. The survivability and structural deformities in zebrafish embryos were assessed by evaluating the coagulation of embryos, lack of somite, and heartbeat following the OECD guidelines 2013 [28].

### 2.4. Imaging of Reactive Oxygen Species (ROS) and Apoptosis in Embryo

Reactive oxygen species (ROS) levels in the embryos injected with CML and different concentrations of SO and OSO were determined by dihydroethidium (DHE) fluorescent staining, as the previously described method [29]. In brief, 20 embryos (5 h post-treatment) were transferred into 24-well plates, washed with water, and stained with 500 μL DHE (final 30 μM). After 30 min incubation in the dark, stained embryos were washed three times with 1 × PBS and visualized under a Nikon Eclipse TE2000 microscope (Tokyo, Japan) at the excitation and emission wavelength of 588 nm and 605 nm, respectively. The extent of apoptosis was measured via acridine orange (AO) staining as a previously described method [30]. Briefly, 20 embryos (5 h post-treatment) were transferred into 24-well plates and suspended with 500 μL of AO (final 5 μg/mL) for 1 h in the dark. After 1 h incubation, stained embryos were washed thrice with 1×PBS and visualized under a Nikon Eclipse TE2000 microscope at 502 nm and 525 nm excitation and emission wavelengths, respectively. The fluorescent intensity from DHE and AO staining corresponding to ROS and apoptosis was quantified using Image J software version 1.53r (http://rsb.info.nih.gov/ij/, accessed on 16 January 2023). 

### 2.5. Acute Inflammation in Adult Hyperlipidemic Zebrafish

Hyperlipidemic zebrafish were prepared by supplementing a 4% cholesterol diet for one month, and acute inflammation was induced by injecting 500 μg CML/10 μL of PBS (equivalent to 6 mM CML considering zebrafish body weight was approximately 300 mg) as per our previous report [31]. Adult zebrafish (n = 210) were randomly divided into six groups (each group n = 35). Zebrafish in Group I were injected with 10 μL PBS only. Zebrafish in Group II were injected with 10 μL PBS containing 500 μg CML. Zebrafish in Group III and IV were co-injected with 500 μg CML with 10 μL of SO (final 1%) and SO (final 2%), respectively. Likewise, zebrafish in Group V and VI were co-injected with 500 μg CML with 10 μL of OSO (final 1%) and OSO (final 2%), respectively. Each treatment was given intraperitoneal (IP) injection using a 28-gauge needle into the abdominal region after anesthetizing zebrafish by drenching in 0.1% solution of 2-phenoxyethanol. At 30 and 60 min post-injection, zebrafish swimming activity and survivability were evaluated. Briefly, swimming activity was analyzed by visualization of tail fin movement and paucity of body paroxysm as previously described method [32], while death was investigated by examining the gill movements, loss of balance, head up or down, drifting on the water surface, or sinking in the bottom of the tank using the OECD guidelines 2019 [33]. After 60 min post-injection Zebrafish in different groups were sacrificed by submerging them in ice slush (~4 °C or less), and the movement of gills was monitored carefully as an indicator of death. The blood from the heart of sacrificed zebrafish was immediately collected using 22-G needle, and hepatic tissue was segregated surgically and processed for histological analysis.

### 2.6. Effect of OSO Supplementation with High-Cholesterol Diet in Adult Zebrafish

The zebrafish (n = 240) were randomly divided into the following four groups (each group n = 60): normal diet control (ND, Tetrabit Gmbh D49304, 47.5% crude protein, 6.5% crude fat, 2.0% crude fiber, 10.5% crude ash, containing vitamin A (29,770 IU/kg), vitamin D3 (1860 IU/kg), vitamin E (200 mg/kg), and vitamin C (137 mg/kg); Melle, Germany), HCD control, HCD with SO (final 20%, *wt*/*wt*), and HCD supplemented with OSO (final 20%, *wt*/*wt*). The 20% supplementation SO and OSO were found optimized and displayed no alteration in swimming activity and mortality of zebrafish (data not shown). Zebrafish were maintained in the abovementioned diet for 6 months at 28°C with a 12:12 h light–dark cycle. Bodyweight and survivability were examined periodically at 3 and 6 months. At the end of 6 months, zebrafish were sacrificed (as a method described in Section 2.5), blood was collected by heart puncture and hepatic tissue was preserved in −70 °C for the histological analysis. 

### 2.7. Analysis of Plasma

Blood (2 μL) was drawn from the hearts of the adult fish, combined with 3 μL of phosphate-buffered saline (PBS)-ethylenediaminetetraacetic acid (EDTA, final concentration, 1 mM) and then collected in EDTA-treated tubes. The plasma total cholesterol (TC) and triglyceride (TG) were determined using commercial assay kits (cholesterol, T-CHO, and TGs, Cleantech TS-S; Wako Pure Chemical, Osaka, Japan) as per the method suggested by the suppliers. In brief, 5 μL serum was mixed with 200 μL reaction mixture (supplied with a commercial assay kit) for the TC analysis. The content was incubated at 37 °C for 10 min, resulting in a red-colored product quantified by adsorption at 490 nm (Microplate reader, Bio-Rad, Hercules, CA, USA). Similarly, 5 μL serum was mixed with a 200 μL TG-specific reaction mixture (supplied with a commercial assay kit) for TG analysis. The content was incubated for 10 min at 37 °C, and the formed colored product was quantified by taking adsorption at 490 nm. For HDL-C analysis, serum was mixed in an equal ratio with the separation solution (supplied with a commercial assay kit), followed by centrifugation at 3000 rpm for 10 min. The supernatant (20 μL) was collected and blended with a 200 μL reaction mixture (supplied with a commercial assay kit). After 10 min incubation at 37 °C, red color intensity corresponding to HDL-C was quantified by taking absorption at 490 nm. Non HDL-C was told by subtracting the value determined for TC from the amount for HDL-C.

### 2.8. Histopathological and Immunohistochemical Analysis

The hepatic tissue was surgically removed from the zebrafish, fixed in 10% formalin buffer solution, and subsequently frozen. The fixed hepatic tissue was embedded in the paraffin, and 7 μm thick slices were prepared and stained with hematoxylin and eosin (H&E) [34].

Oil red O staining was performed as the previously described method [31]. In brief, 7 μm thick hepatic tissue sections were prepared and mounted on the slide. The tissue section was stained with oil red O solution for 15 min at 60 °C, followed by washing with 60% isopropanol. The section was stained with hematoxylin for 30 s, washed with water, and visualized under a microscope. 

Immunohistochemical (IHC) staining was performed to detect proinflammatory cytokine IL-6 as the previously described method [35]. In brief, IHC primary antibody (ab9324, Abcam, London, UK) was diluted (1:200) according to the manufacturer’s instructions and incubated overnight at 4 °C with tissue section. The IHC reaction was visualized using the EnVision + System-HRP polymer kit as a secondary antibody (1:1000, Code K4001, Dako, Glostrup, Denmark). The tissue was visualized under an optical microscope (Nikon, Tokyo, Japan). Hepatic DHE staining for the ROS analysis was performed using the same protocol described in Section 2.4.

### 2.9. Statistical Analysis

Data are presented as mean ± standard deviation of three independent experiments. The statistical difference between the groups was determined by one-way analysis of variance (ANOVA) followed by Tukey’s multiple-range test at *p* < 0.05 using Statistical Package for the Social Sciences software program (version 23.0; SPSS, Inc., Chicago, IL, USA).

## 3. Results

### 3.1. OSO Attenuated the CML-Induced Embryo Toxicity

The survival rate of zebrafish embryos treated with CML and different concentrations of SO and OSO is depicted in Figure 1A. The PBS-alone-injected group showed the highest survivability (82.6%). While the CML (500 ng)-injected embryo survivability decreased periodically and reached 28.6% at 24 h, in contrast, improved embryo survivability was observed in the embryos co-treated with the CML and OSO1% or OSO2% as compared to only CML-treated embryos. Likewise, embryos co-treated with CML with SO1% and SO2% also displayed improved survivability compared to only CML-treated embryos. A significantly higher embryo survivability (*p* < 0.05), around 2.1-fold (60.6%) and 1.8-fold (49.9%), was observed in OSO2%- and OSO1%-injected embryos, respectively, as compared to only CML-injected embryos (28.6%) at 24 h.

Morphological changes in the embryo’s development were observed at 5 h and 24 h post-treatment (Figure 1B). A delayed embryo development with structural deformities was observed in CML-injected embryos compared to only PBS-injected embryos. Most of the embryos (96%) recovered from the CML-injected group displayed severe deformities of the eyes and tail development with abnormal heart rate. Similarly, most embryos (87%) recovered from the SO1% group displayed deformities regarding eyes and tail development and abnormal heart rate comparable with only CML-injected embryos. In contrast to the SO1% group, most of the embryos (81%) in the SO2%-treated group displayed resistance against developmental deformities posed by CML. Convincingly, the embryos co-injected with OSO1% and OSO2% revert the CML-induced developmental impairment, and the results are in accordance with findings of embryo survivability (Figure 1A). Only 11% and 7% of embryos recovered from the OSO1% and OSO2% groups showed minor developmental deformities. These findings demonstrated that OSO, mainly OSO2%, has strong protective activity against CML-induced developmental deformities and mortality of embryos zebrafish.

Fluorescence intensity (FI) from AO and DHE staining indicated that apoptosis severe ROS production occurred in embryos, respectively, by the presence of CML (Figure 1C). AO staining revealed the highest FI in CML-treated embryos, which was approximately 3-fold higher than PBS-treated embryos, thus manifesting the apoptotic potential of CML. In contrast, the OSO1% and OSO2% treatments significantly (*p* < 0.05) rescued the embryos from CML-induced apoptosis. A 47.2% and 59.1% less AO fluorescent intensity was noticed in OSO1%- and OSO2%-injected embryos compared to only CML-injected embryos. Additionally, SO1%- and SO2%-injected embryos displayed significant (*p* < 0.05) apoptotic inhibition evident by 33.8% and 44.1%, respectively, reduced fluorescent intensity compared to CML-injected embryos. 

The results of DHE staining showed a massive production of ROS in response to CML, which was significantly 4.2-fold higher than only PBS-treated embryos. The CML-induced ROS production was significantly reverted by injecting OSO and SO. The embryos injected with OSO1% and OSO2% displayed 58.3% and 70.2% reduced DHE fluorescent intensity compared to CML-treated embryos. Additionally, a 41.6% and 54.1% reduced DHE fluorescent intensity was noticed in embryos treated with SO1% and SO2%, respectively, compared to only CML-treated embryos. The apoptosis and ROS staining results imply the significant (*p* < 0.05) efficacy of SO and OSO at both 1% and 2% concentration to preclude CML-induced apoptosis and ROS production.

### 3.2. OSO Rescued HCD-Fed Zebrafish from CML-Induced Mortality and Acute Paralysis

As shown in Figure 2A, severe mortality with only 17.5% survival of zebrafish was noticed in the CML alone group compared to this 100% survivability in PBS alone group, signifying the severe toxicity of CML. A non-significant effect of the treatment of SO1%, SO2%, and OSO1% was observed against CML-induced mortality in zebrafish. However, treatment of OSO2% significantly (*p* < 0.05) countered the CML triggered mortality in zebrafishes. The OSO2% group showed around 3.7-fold and 2.0-fold higher survivability than the CML-alone and SO2% groups, respectively, at 60 min post injection. As shown in Figure 2B, the CML-alone group and SO1%, SO2%, and OSO1% groups showed severe CML-induced paralysis after 30 min of post-injection, where almost all the zebrafish were lying on the bottom of the tank without any swimming activity (Figure 2C, Appendix A). However, zebrafish co-injected with OSO2% significantly (*p* < 0.05) recovered from the paralytic effect of CML. A 32.5% restoration of zebrafish swimming activity was noticed at 30 min post-treatment of OSO2%, which was linearly increased up to 52.5% at 60 min of post-treatment, representing 13- and 3-fold higher recovery as compared to only CML-treated zebrafish at the respective time. Additionally, a 2-fold higher zebrafish swimming activity was noticed in the OSO2% group compared to the SO2% injected group at 60 min post-treatment. The results demonstrated the preventive role of OSO2% by the restoration of swimming activity against CML-induced acute paralysis in zebrafish that consequently improved survivability (Appendix A).

### 3.3. OSO Ameliorated the CML-Induced Hepatic Inflammation in HCD-Fed Zebrafish

As shown in Figure 3A, of the H&E staining, a visible hepatic degeneration around the portal vein (as indicated by black arrows) was observed in CML and CML co-injected with SO1%. Compared to SO1%, SO2% showed a slightly better recovery of CML-induced hepatic degeneration. In contrast, OSO efficiently subdued the CML-induced hepatic degeneration visible by the low stained area of the nucleus. The hepatic tissue of the OSO1% and OSO2% groups displayed 29.9% and 49.9% lower H&E-stained areas than the hepatic tissue of only the CML-injected group. A significantly reduced around 34.3% (*p* = 0.001) reduction of the H&E-stained area in the OSO2%-injected group compared to the SO2%-injected group clearly indicates the efficient hepatoprotective nature of OSO2%.

As shown in Figure 3B, the higher oil red O-stained area corresponding to fatty liver changes was observed in the hepatic tissue of CML-injected zebrafish, which was significantly reduced with the treatment of SO and OSO. The zebrafish co-treated with SO1% and SO2% revealed 23.8% and 34.7% lower stained areas than the CML-alone group. In contrast to this, a 50.4% (*p* < 0.001) and 81.5% (*p* < 0.001) lower oil red O-stained area was observed in the OSO1% and OSO2% groups, respectively, compared to the CML-treated group. While compared with SO2%, a significantly 71.6% (*p* < 0.001) lower oil red O-stained area was observed in the OSO2% injected group. Interestingly, zebrafish in SO and OSO groups (at both 1% and 2%) displayed a much lower oil red O-stained area as compared to only PBS-injected groups, signifying the potential of SO and OSO to prevent fatty liver changes. However, the most significant difference with 78.2% (*p* < 0.001) lower oil red stained area was observed in SO2% compared to the only PBS-injected group. 

DHE staining demonstrated the higher production of ROS in the CML-injected zebrafish, apparent by the 74.3% red-stained area (Figure 3C). Compared to this, a slight reduction of the DHE-stained area was noticed in the hepatic tissue fetched from SO2% injected zebrafish. A 61.7% DHE stained area was quantified in SO2% injected zebrafish, which was 1.2-fold lower than that of the only CML-treated group. However, injection of OSO precisely OSO2% effectively reduces the ROS production as demonstrated by only 38.3% red-stained area, which is almost 1.9-fold lower than that of the stained area manifested in the only CML-treated group. Compared to SO2%, a 1.6-fold lower DHE-stained area appeared in the OSO2% injected group, signifying the higher efficacy of OSO2% to counter ROS.

Furthermore, IL-6 production in different groups was examined by the IHC staining. As depicted in Figure 3D, 37.2% IHC stained area corresponding to the IL-6 production was noticed in the hepatic tissue of the PBS-injected and CML-injected groups, respectively. Injection of SO1%, SO2% and OSO1% did not affect the IHC-stained area, evidenced by 33.5%, 36.6%, and 35.1% stained areas, respectively, which is nearly similar to the stained area (37.2%) corresponding to the CML alone injected group. In contrast, the OSO2% injected group efficiently reduced IL-6 production. A 25.5% IHC stained area was noticed in the OSO2% injected group, which was approximately 1.5-fold lower than that of the stained area that appeared in the CML-treated group. Compared to SO2%, a significantly reduced IHC stained area around 1.4-fold (*p* = 0.044) in the OSO2% injected group testifies to the superiority of OSO2% to block IL-6 production in zebrafish. The staining results collectively suggest that OSO2% efficiently reverted the CML-induced adverse effect exerted by ROS and proinflammatory IL-6 production. 

### 3.4. OSO Attenuated CML induced Dyslipidemia in HCD-Fed Zebrafish

Plasma lipid profiles of the zebrafish injected with CML and subsequently treated with SO and OSO are demonstrated in Figure 4. The least HDL-C level was detected in the CML-treated group. The treatment of SO and OSO at the tested concentration of 1% and 2% significantly (*p* < 0.05) enhanced the HDL-C compared to the only CML-injected group. A 29.7% and 20.7% higher HDL-C was detected in the zebrafish injected with SO2% and OSO2%, respectively, compared to the only CML-treated zebrafish. Contrary to HDL-C, a higher amount of TC was quantified in the CML-injected group, which was significantly (*p* < 0.05) reduced in response to SO and OSO at both 1% and 2% concentrations. Furthermore, a reduced HDL-C/TC ratio was noticed in the CML-injected zebrafish that was significantly (*p* < 0.05) improved by the injection of both OS and OSO. The treatment of SO1% and SO2% enhanced the HDL-C/TC ratio by 21.2% and 23.9%, respectively, compared to the CML-injected group. However, a more profound effect with 31.4% and 38.5% enhancement in the HDL-C/TC ratio was observed in the OSO1%- and OSO2%-injected group, respectively, compared to the only CML-injected group. Furthermore, the OS and OSO injection significantly (*p* < 0.05) reduced CML-induced TG levels. An 18.6%, 24.9%, 49.9%, and 53.1% lower TG level was quantified in the SO1%-, OS2%-, OSO1%-, and OSO2%-injected groups, respectively, as compared to the only CML injected groups. Likewise, a massive level of non-HDL-C was confirmed in zebrafish injected with CML that was convincingly deterred by injecting SO and OSO. A 1.6- and 1.7-fold lower non-HDL-C was quantified in SO1%- and SO2%-infused groups compared to the only CML-injected group. In contrast to the SO treatment, a more intense effect with 2.6- and 3.6-fold reduction in non-HDL-C level was demonstrated by OSO1% and OSO2%, respectively, compared to the only CML-injected group. The results strongly advised the effect of SO and OSO on balancing the plasma lipid profile in the hypolipidemic zebrafish.

### 3.5. OSO Supplementation Increased Survivability and Decreased Body Weight of HCD-Fed Zebrafish

The survivability of zebrafish fed with HCD and the supplementation of HCD with SO and OSO were monitored up to 6 months, and the outcomes are presented in Figure 5. After 3 months of feeding, the ND group showed 100% survivability, whereas the survivability reduced to 84.0% in the HCD-fed group. In contrast, 96.7% survivability was observed in the HCD + OSO-supplemented group. Surprisingly, the HCD + SO-fed group exhibited the most negligible effect, with a 75.0% survivability after 3 months of feeding. Further, with an enhancement of feeding time up to 6 months, survivability was reduced in all the groups. However, better survivability was observed in HCD + OSO-fed groups compared to only HCD- and HCD + SO-fed groups.

Additionally, a change in zebrafish body weight was observed after 3 and 6 months of feeding. All the groups that received the different food showed an enhancement in their body weight after 3 months of feeding compared to their body weight at the beginning of the experiment. However, the most noticeable effect with 61.8% and 31.3% body weight enhancement was observed in the HCD + SO- and only HCD-fed groups, respectively, after 3 months compared to their initial body weight. Contrary to this, the least body weight enhancement, with 8.5% and 18.9%, was observed in the ND- and HCD + OSO-supplemented groups, respectively. Further enhancement of the feeding duration up to 6 months had a non-significant impact on the body weight gain in the zebrafish that received ND and HCD + OSO diet. However, a slight enhancement (5%) in body weight was observed in the HCD + OS-fed group compared to the body weight observed at 3 months. Unlike this, a colossal change with 49.5% advancement in body weight was observed in the HCD-fed zebrafish after 6 months compared to their body weight of 3 months.

After three months’ consumption, the body weight of zebrafish in the ND- and OSO + HCD-fed groups were 20.3% and 12.2% lower than that of zebrafish in the HCD-control group. Contrary to this, a 16.2% body weight enhancement was observed in the SO + HCD-fed group compared to the HCD-alone group. At six months after consumption, 46.4%, 41.2% and 18.4% lower body weights were observed in ND-, OSO + HCD- and SO + HCD-fed groups, respectively, compared with only the HCD group. Interestingly, non-significant body weight changes between ND- and OSO + HCD-fed groups were observed during the six months of feeding, signifying the preventive role of OSO in maintaining body weight altered by HCD.

Plasma lipid profile analysis revealed that HCD increased plasma TG and TC levels by 1.7 and 3.1 folds, respectively, compared to the ND group (Figure 5C). However, OSO supplementation effectively prevents HCD-enhanced plasma TG and TC levels. Surprisingly, a significantly (*p* < 0.05) higher TC level was observed in SO supplementation with HCD compared to only HCD groups. However, SO supplementation significantly (*p* < 0.05) reduced HCD-induced TC. The results supported the anti-obesity effect of OSO.

### 3.6. OSO Supplementation Suppressed the HCD Provoked Hepatic Steatosis and ROS Production

Hepatic histology is shown in Figure 6. Histological analysis with H&E staining indicated that the stained area of the nucleus in the HCD group significantly increased by 10% compared with that in the ND group. OSO supplementation significantly (*p* < 0.05) decreased the stained area of the nucleus by 5% compared with the HCD control group (Figure 6A).

Consistent with H&E results, the hepatic oil red O-stained area in the HCD-fed group was significantly (*p* < 0.05) 3-fold higher than that of the ND group, whereas OSO supplementation with HCD significantly (*p* < 0.05) decreased the hepatic oil red O-stained area and was similar to that of the oil red O-stained area of ND group (Figure 6B). Furthermore, the DHE-stained area corresponding to ROS production was significantly 2.4-fold higher in the HCD control group compared with that in the ND group. The OSO supplementation with HCD significantly repressed the DHE-stained area compared with the HCD control and SO-supplemented HCD group (Figure 6C). These findings strongly imply the impact of OSO against HCD-induced hepatic steatosis.

## 4. Discussion

Abnormal lipid profile, oxidative stress, and inflammation are severe pathological components, responsible for the onset of several diseases, such as heart failure, diabetes, and dementia [36,37]. Being overweight and obese is also a global challenge that enhances the risk of various disorders such as heart disease [38]. Much research and development is underway to develop efficient, safe, natural compounds, oils and extracts that can maintain the lipid profile, oxidative stress, and inflammation.

Sunflower oil (SO) is the second most widely used edible oil and has a substantial derma protective and antimicrobial effect [39]. Medicinal and functional properties of SO can be improved by ozonation [40]. 

Accumulating reports revealed the OSO displayed potent antioxidant and antimicrobial potential [23,41,42]; nonetheless, a limited study revealed the other beneficial effects of OSO precisely against dyslipidemia and hepatic inflammation in zebrafish [23]. However, our research group has been working on different bio-functionalities of OSO; in one such effort, we have confined the protective effect of OSO against oxidative stress-induced damage in zebrafish embryos [23]. To extend our study herein, we attempted to decode the comparative effect of OSO and SO against CML-induced toxicity and hepatic inflammation in HCD-fed zebrafish. 

CML is well-known for its toxicity owing to the induction of ROS/oxidative stress [31,43] that provokes the cellular mechanism of apoptosis. A provocative role of CML on the proapoptotic regulators such as Bcl-2/Bax, caspase 3, and caspase 9 has also been documented [44]. In the present study, CML was used as an inducer for oxidative stress and inflammation in the hyperlipidemic zebrafish as our recent report [27] and the amelioration effect of SO and OSO was evaluated. We observed that OSO effectively protected the embryo survival and restoration of structural and development deformities against the adversity imposed by CML.

Moreover, an enhanced effect was exhibited by OSO2% compared to the SO2%, suggesting the positive impact of ozonation on the functionality of SO. These observations are in accordance with the previous study that documented the merits of ozonation over native oils [23]. The superior activity of OSO2% was probably due to the presence of a variety of ozone-catalyzed compounds [22,40] that are well-known for their diverse biological activities [45]. Furthermore, OSO2%, as compared to SO2%, strongly diminishes the CML-induced oxidative stress evident by the low fluorescent intensity in DHE-stained embryos. Additionally, the low intensity of AO-stained green fluorescence in embryos injected with CML + OSO2% compared to only CML and CML co-injected with SO2% demonstrates inhibition of apoptosis by OSO2%. The outcomes of AO staining endorse the results observed for embryo survivability and strengthen the notion that OSO2% promotes anti-apoptotic events, consequently leading to higher embryo survival. Numerous reports suggest that ROS-induced oxidative stress provokes the apoptotic cascade, leading to cellular death [46,47].

We presume that OSO, owing to its significant antioxidant activity [23], scavenges ROS or induces the cellular antioxidant machinery, renders the CML-induced oxidative stress, and finally prevents embryo death. Additionally, an affirmative role of OSO in the structural alteration of HDL_3_ has been described [23]. The altered HDL_3_ enhances the functionality of paraoxonase-1, a vital biomarker associated with counter inflammation and oxidative stress [48].

The finding collectively indicates that OSO2% due to its antioxidant nature balances the CML-induced oxidative stress and subsequently prevents the inflammation and apoptosis and suppresses the mortality and developmental deformities of zebrafish embryos (Figure 7). Furthermore, the effect of OSO against CML-induced acute paralysis was determined in adult zebrafish fed with HCD (4%). In our previous study, we documented the resistance of HCD-fed zebrafish towards the low dose of CML [31], hence, we have treated the zebrafish with the high amount of CML (final 500 μM, equivalent to 6 mM in around 300 mg body weight of zebrafish) and checked the acute paralysis by observing the swimming behavior at 30 and 60 min post-injection [27]. The OSO2% group showed a great resilience against the acute paralytic toxicity of the CML evident by the higher swimming percentage (32.5%), which progressively increased up to 52.5% with time. In contrast, SO1% and SO2% were found to be less effective towards the revival of swimming activity against CML-induced acute paralysis. However, the exact mechanism of survival against the CML-induced paralysis is not known. The antioxidant and anti-inflammatory role of OSO may be a leading cause for such an effect. This notion is supported by reports showing the association of inflammation with paralytic disorder [27,31]. Even more, our recent study described the effective role of CIGB-258 and tocilizumab (IL-6 inhibitor) towards the revival of zebrafish swimming patterns against CML-induced acute paralysis [27,31]. Furthermore, we have reported that CIGB-258 inhibited the IL-6 production and thus suppresses the inflammation, implying that inflammation may be one culprit for acute paralysis [30].

Additionally, OSO displayed hepatic prevention against CML-induced hepatic damage in HCD-fed zebrafish. A higher infiltration of neutrophils, as depicted in the H&E-stained area, was dose-dependently reduced by the treatment of OSO. The Oil red O staining suggested the curative role of OSO2% against CML-induced fatty liver changes in hyperlipidemic adult zebrafish. Consistent with this, the OSO2% diminished a CML-induced higher production of IL-6 and ROS in HCD-fed zebrafish. IL-6 is an important proinflammatory cytokine associated with various disorders such as rheumatoid arthritis (RA), Crohn’s disease, and multiple sclerosis [49]. An antioxidant role of OSO2% might be a reason for the inhibition of IL-6, consistent with the reports suggesting an inflammatory pathway induction by oxidative stress [50]. The combined results of ROS staining in embryos and of the hyperlipidemic zebrafish suggest the effect of OSO2% to balance oxidative stress, which is consistent with our earlier report suggesting an antioxidant nature of OSO [23].

Although several studies have reported that OSO has antioxidant, antimicrobial, and antifungal activities, only a few studies have evaluated its lipid-lowering effects. Herein, OSO2% displayed the healing effect on dyslipidemia by reducing TG, TC, and non-HDL-C levels in HCD-supplemented CML-injected zebrafish. Elevated IL-6 might be the main culprit of the lipid profile imbalance in the CML-treated group. The notion was strongly supported by the previous findings demonstrating a correlation between serum IL-6 and TG [51]. Additionally, the previous report suggested that IL-6 stimulated the TG secretion in the blood of rats just after 2 h post-injection [52]. Our findings demonstrated the low accumulation of IL-6 in the hepatic tissue treated with OSO2% and consequently have a balancing effect on the serum lipid profile. Varieties of the inflammatory disorder emerge with the imbalance of the serum lipid profile, such as in the patients of RA, a decreased HDL-C and apolipoprotein A-I level with an elevation of IL-6 were observed [53,54].

Similarly, in systemic lupus erythematous (SLE) and psoriasis, a decreased HDL and higher TG levels were observed [55]. Moreover, many drugs for treating RA, SLE, and psoriasis significantly affect the serum lipid profile, thus strongly advocating the central role of inflammation in balancing the lipid profile [55]. The pathophysiology of infection and inflammation suggested the deep involvement of cytokines, including IL-6, in the abnormal serum lipid profile [56]. For example, several cytokines, such as IL-6, were found to actively increase serum TG and VLDL [56]. We believe the anti-inflammatory activity (suggested by IL-6 staining) is the major reason responsible for the balancing of serum lipid profile by OSO2%.

A noteworthy role of OSO was also observed in maintaining the body weight, liver fibrosis and lipidosis in zebrafish against prolonged consumption exposure to HCD. We speculated these effects originated from the antioxidant and anti-inflammatory activity of OSO, which following the published article suggesting an association between obesity and oxidative stress, lipid metabolism, and IL-6 [37,57].

## 5. Conclusions

OSO neutralizes CML-induced toxicity by suppressing oxidative stress and apoptosis, consequently preventing zebrafish embryos’ developmental deformities and mortality. The OSO treatment triggered a prompt revival of zebrafish from CML-induced paralysis with survivability from acute hepatic inflammation and displayed a significant credential to maintain dyslipidemia and steatosis. OSO prevents the fatal effect exerted by prolonged intake of HCD by maintaining hepatic morphology, serum lipid profile, body weight maintenance and higher survivability. The results endorse OSO as a natural curative agent for hyperlipidemia, oxidative stress, and inflammatory-related disorders that could be utilized commercially to deal with such ailments after sufficient clinical trials. 

## Figures and Tables

**Figure 1 antioxidants-12-01240-f001:**
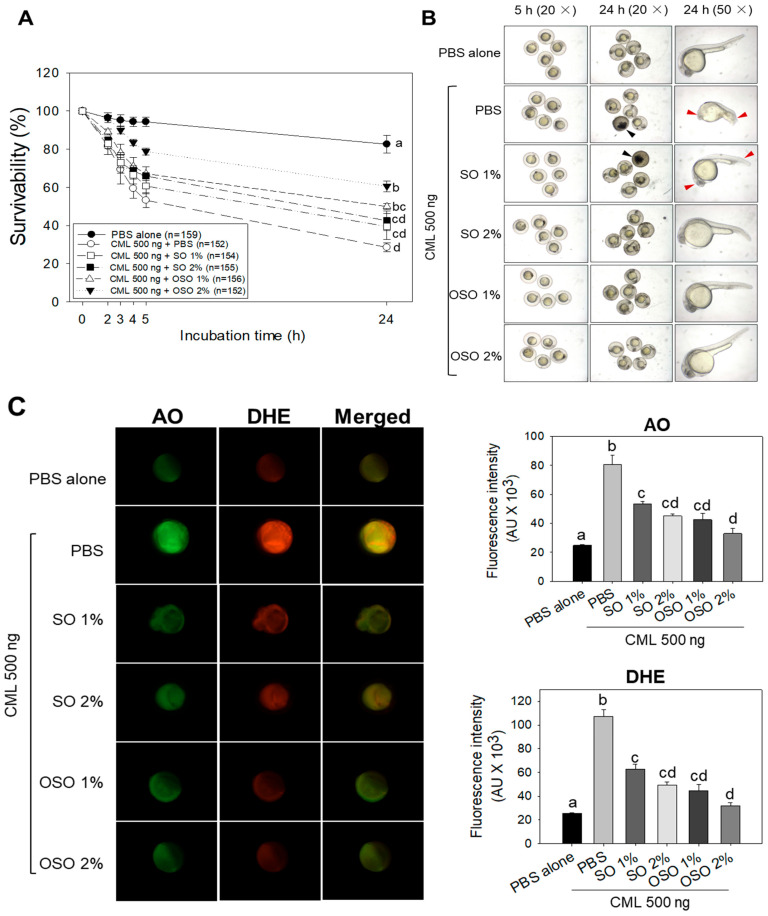
Effects of sunflower oil (SO) and ozonated sunflower oil (OSO) on carboxymethyllysine (CML)-induced toxicity in zebrafish embryos. (**A**) Survivability of zebrafish embryos during 24 h post-injection; (**B**) stereo images of developing zebrafish embryos at 5 h and 24 h post-treatment. The black arrow represents the death of embryos during 5 h post-injection, while the red arrow represents developmental deformities (of eyes and tail curvature) and stunted growth at 24 h post-injection. (**C**) Fluorescent images of acridine orange (AO) and dihydroethidium (DHE) represent the apoptosis and ROS production in zebrafish embryos at 5 h post-injection. PBS alone group received microinjection of PBS, while the PBS + CML group received the microinjection of 500 ng CML in PBS; the CML + SO1% and SO2% groups were microinjected with 500 ng CML with SO1% and SO2%, respectively; similarly, CML + OSO1% and OSO2% groups were microinjected with 500 ng CML with OSO1% and OSO2%, respectively. The fluorescent intensity of AO and DHE stained area was quantified by Image J software version 1.53r (http://rsb.info.nih.gov/ij/ accessed on 23 January 2023). Letters (a–d) represent the statistical significance (*p* < 0.05) between CML alone and CML + OS or OSO-injected groups.

**Figure 2 antioxidants-12-01240-f002:**
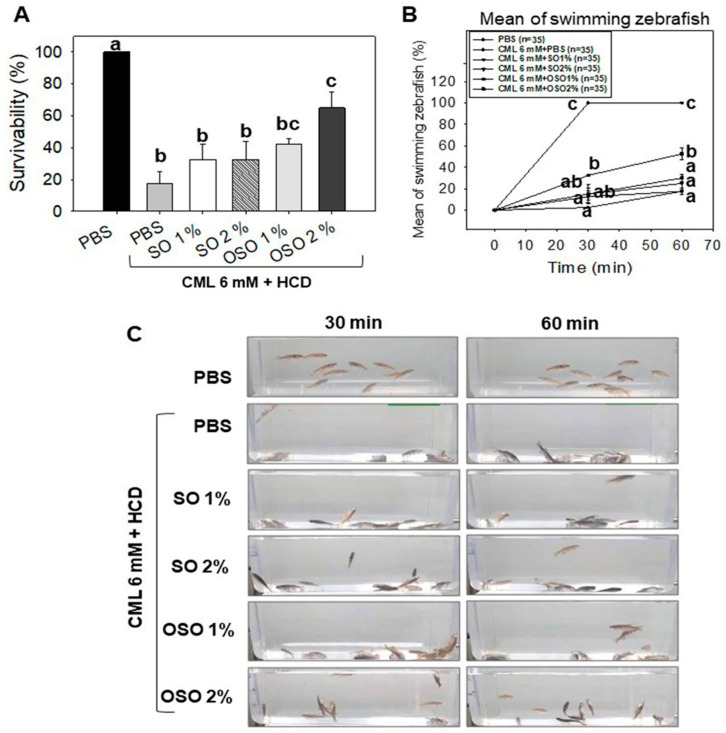
Effects of sunflower oil (SO) and ozonated sunflower oil (OSO) on survivability and swimming ability of carboxymethyllysine (CML) injected adult zebrafishes maintained on the high-cholesterol diet (HCD). (**A**) Survivability of adult zebrafish after 60 min post-injection of PBS alone or CML (500 μg equivalent to 6 mM) or CML co-injected with SO1% or SO2% and OSO1% or OSO2%. (**B**,**C**) Swimming behavior of adult zebrafish measured at 30 min and 60 min post-injection of PBS alone or CML (500 μg equivalent to 6 mM) alone or CML in the presence of SO1% or SO2% and OSO1% or OSO2%. Letters (a–c) represent the statistical significance (*p* < 0.05) between the groups.

**Figure 3 antioxidants-12-01240-f003:**
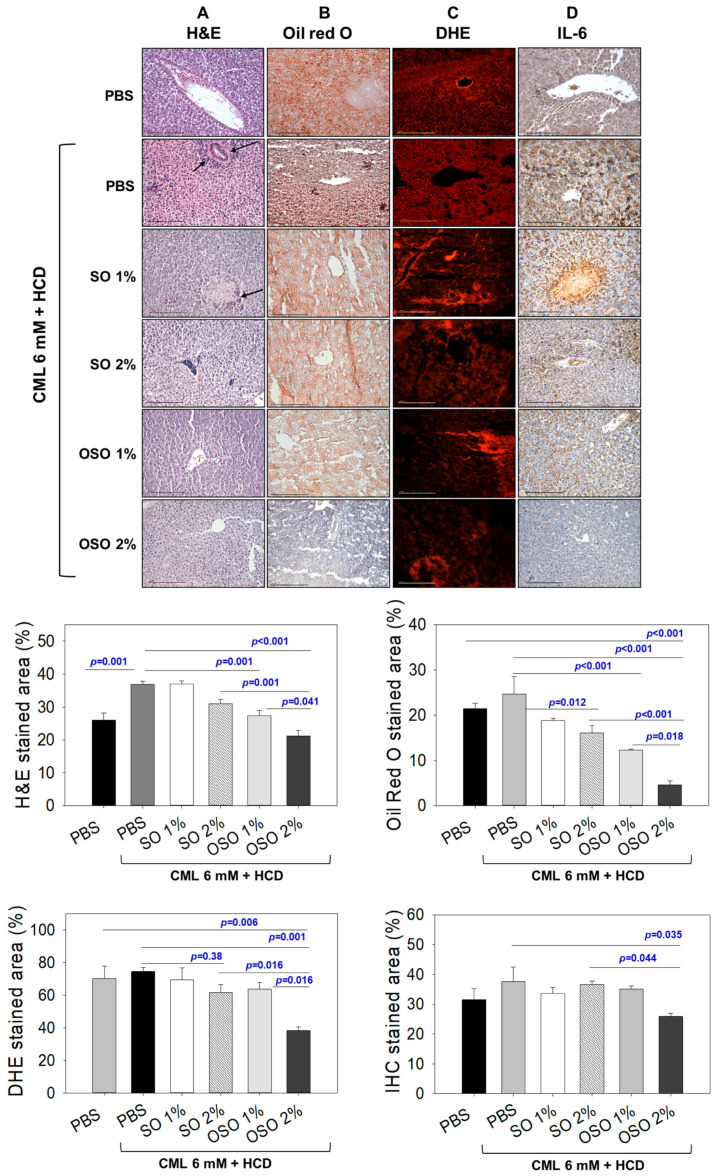
Hepatic morphology, fatty liver changes, reactive oxygen species (ROS), and IL-6 production in high-cholesterol diet (HCD) consumed adult zebrafish injected with carboxymethyllysine (CML) or CML with sunflower oil (SO) and ozonated sunflower oil (OSO). (**A**) Hematoxylin and eosin (H&E) staining. (**B**) Oil red O staining. (**C**) Dihydroethidium (DHE) fluorescent staining for the detection of ROS. (**D**) Immunohistochemistry (IHC) for the analysis of IL-6 production. Scale bar equivalent to 100 μm. PBS alone group received microinjection of PBS, PBS + CML group received the microinjection of CML (500 μg equivalent to 6 mM) in PBS, while the CML + SO1% and SO2% groups were microinjected with 500 μg of CML with SO1% and SO2%, respectively; similarly, the OSO1% and OSO2% groups were micro-injected with 500 μg of CML with OSO1% and OSO2%, respectively. Zebrafish received a high-cholesterol diet (HCD) during the experiment. All the stainings were performed on the hepatic tissue surgically acquired at 60 min post-injection. All the images were captured under a microscope at 400× magnification. The stained area in all the histological analyses was appraised by Image J software version 1.53r (http://rsb.info.nih.gov/ij/ accessed on 30 January 2023).

**Figure 4 antioxidants-12-01240-f004:**
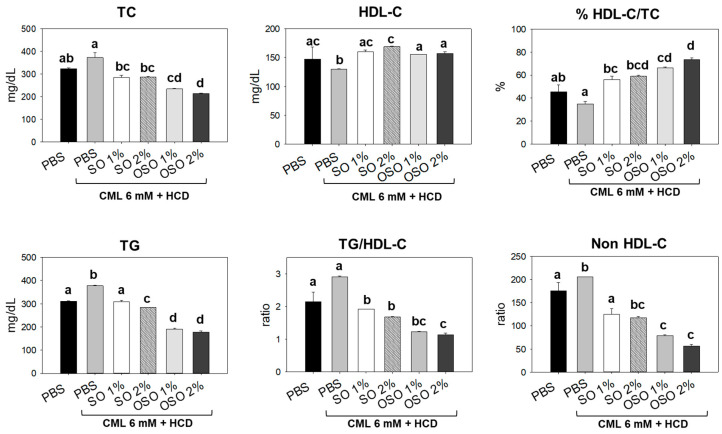
Effects of sunflower oil (SO) and ozonated sunflower oil (OSO) on plasma lipid profiles in carboxymethyllysine (CML) injected adult zebrafish maintained on the high-cholesterol diet (HCD). Blood for the lipid profile from adult zebrafish was collected at 60 min post-injection. PBS alone group received microinjection of PBS, PBS + CML group received the microinjection of CML (500 μg equivalent to 6 mM) in PBS, while the CML + SO1% and SO2% groups were microinjected with 500 μg of CML with SO1% and SO2%, respectively; similarly, the CML + OSO1% and OSO2% groups were microinjected with 500 μg of CML with OSO1% and OSO2%, respectively. Alphabets (a–d) above the bar graphs represent the statistical significance (*p* < 0.05) between different groups. TG, TC, and HDL-C are abbreviated for triglyceride, total cholesterol, and high-density lipoprotein cholesterol, respectively.

**Figure 5 antioxidants-12-01240-f005:**
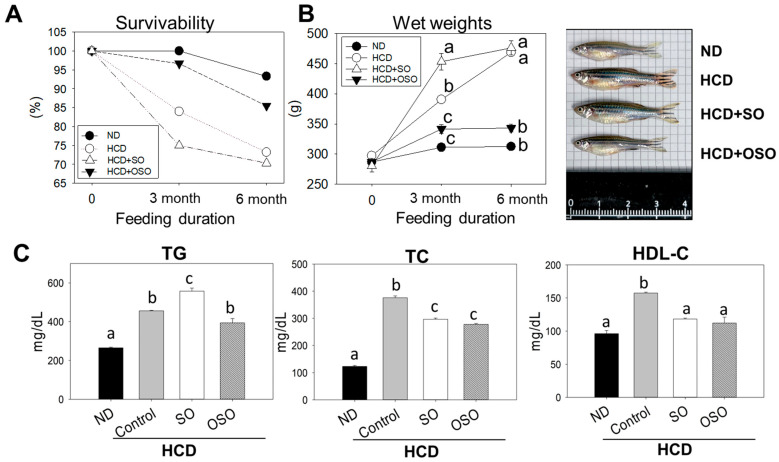
Effects of sunflower oil (SO) and ozonated sunflower oil (OSO) on the survivability, body weight, and lipid profile of zebrafish supplemented with high cholesterol diet (HCD). (**A**) survivability of zebrafish during 6 months of HCD feeding. (**B**) Changes in the body weight of zebrafish during 6 months of HCD feeding. (**C**) Plasma lipid profile at 6 months in HCD-fed zebrafish. ND, normal diet; HCD, high-cholesterol diet (4% cholesterol blended with ND); HCD + SO (HCD with SO (final 20%, *wt*/*wt*)); HCD + OSO (HCD with OSO (final 20%, *wt/wt*)). Different letters (a–c) above the bar graphs represent the statistical significance (*p* < 0.05) among different groups. TG and TC are abbreviated for triglyceride and total cholesterol, respectively.

**Figure 6 antioxidants-12-01240-f006:**
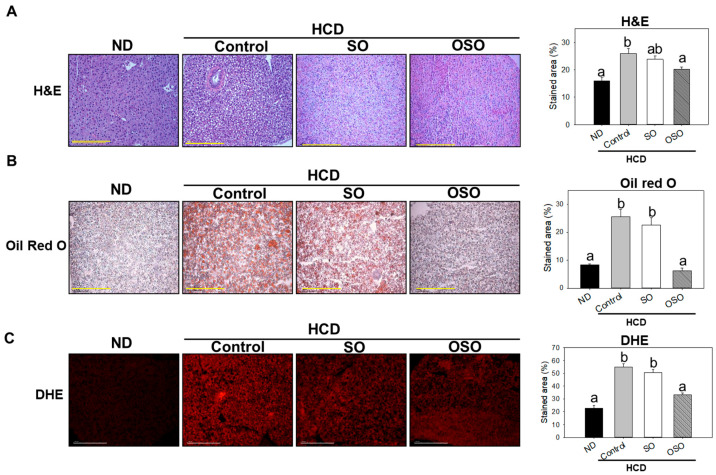
Assessment of hepatic morphology, fatty liver changes, and reactive oxygen species (ROS) in adult zebrafish fed with high-cholesterol diet (HCD) up to 6 months. (**A**) Hematoxylin and eosin (H&E) staining. (**B**) Oil red O staining. (**C**) Dihydroethidium (DHE) ROS examination. Scale bar equivalent to 100 μm. All the images were captured under a microscope at 400× magnification. The stained area of the histology was assessed by Image J software version 1.53r (http://rsb.info.nih.gov/ij/ accessed on 8 January 2023). Letters (a,b) above the bar graphs represent the statistical significance (*p* < 0.05) between the groups. ND, normal diet; HCD, high-cholesterol diet (4% cholesterol blended with ND); HCD + OSO (HCD with SO (final 20%, *wt*/*wt*)); HCD + OSO (HCD with OSO (final 20%, *wt*/*wt*)).

**Figure 7 antioxidants-12-01240-f007:**
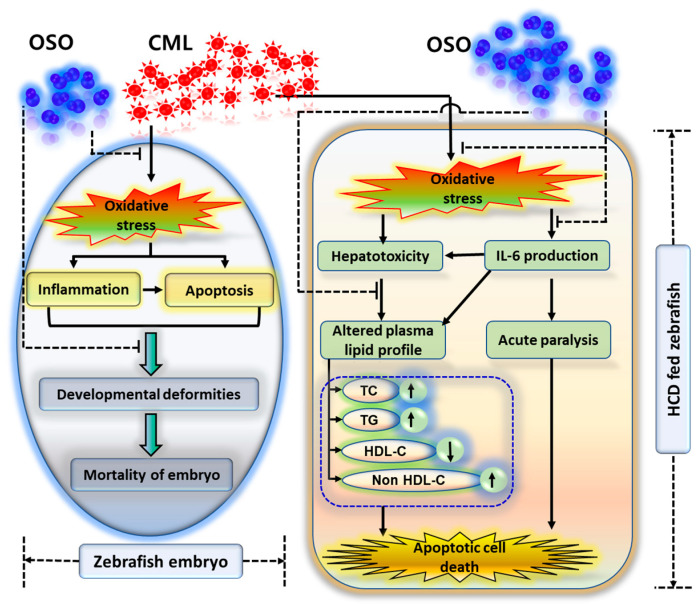
Ozonated sunflower oil (OSO) mediated preventive events against carboxymethyllysine (CML) induced toxicity in zebrafish embryos and hyperlipidemic zebrafish.

## Data Availability

The data used to support the findings of this study are available from the corresponding author upon reasonable request.

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
