# Peer review of "Long-Term Supplementation of Ozonated Sunflower Oil Improves Dyslipidemia and Hepatic Inflammation in Hyperlipidemic Zebrafish: Suppression of Oxidative Stress and Inflammation against Carboxymethyllysine Toxicity"

_antioxidants, 2023, doi:10.3390/antiox12061240_

Round 1

Reviewer 1 Report

This paper reports on the effects of ozonated sunflower oil (OSO) against dyslipidaemia and hepatic inflammation in hyperlipidaemic zebrafish. The experimental design consisted of three different exposures in different life stages (embryo and adult). The results show OSO to mediate preventive events against carboxylmethyllysine-induced toxicity in zebrafish embryos and also against hyperlipidemic zebrafish.

In general, this is an interesting study describing beneficial effects for OSO supplementation in zebrafish which could led to the development of new formulations for aquaculture. Yet, some issues need to be addressed:

- The introduction needs to be revised to include further connection between ideas. For instance, authors describe dyslipidemia and its relationship with oxidative stress and proinflammatory signaling pathways. Yet, nothing is described regarding the ozone effects on these parameters. What is already know about ozone, oxidative stress, inflammatory signals and zebrafish?

- L73, why three different experiments were conducted and how do they relate with the objective/hypothesis of the work?

- L91, which “slight modifications” were performed?

- L95, how many embryos and replicates were done?

- L99, why were these concentrations of OSO tested here?

- L106, overall, the methods are only slightly described with too many details missing. For instance, ROS and apoptosis were observed in embryos at which time-point?

- L118, is this one-month treatment enough to cause dyslipidaemia?

- L128, how was behaviour assessed? Which parameters were analysed?

- L129, how were animals sacrificed?

- L133, were 60 animals used for each group? Or were 15 animals used (n=4)?

- L164, parametric data should be described as mean and SD not SE and non-parametric as median and interquartile ranges. In addition, before applying parametric tests, normality should be tested.

- L172, is the n reported correct? Why so many embryos were used (n=156)?

- L172, why different nomenclatures were used for statistics (* and letters)? It is better to only use letters to denote statistical differences between groups.

- L181, morphological abnormalities should be quantified.

- L232, what was the survivability of PBS group? In fact, this group (negative control) is missing in the analysis and should be included to show that CML induce the desired effects.

- L421-452, these sentences should be reduced or eliminated.

- L458, it seems that OSO1% also reverted some effects from CML.

- L464, was the ozone-catalyzed compounds assessed in this study to support this sentence?

- L475-482, this is too speculative. In fact, how do the concentrations relate with the doses applied in the rat? Furthermore, the administration was different (intragastrically vs intraperitoneal).

- L496, “probably” should be removed.

- L497, which reports show this association?

Author Response

Thank you very much heartily for your valuable review and comments to improve this paper

Please find attached doc as point-to-point response 

Reviewer 2 Report

The Manuscript covers interesting topics and is well organized. However, important revisions are needed, such as the introduction of the control group in one of the experiments and important explanations on the choice of SO and OSO concentration used in chronic treatments.

Major revisions:

- Why is there no control group (only injected with PBS) for IP injection experiments on adult zebrafish?!

- Why did they use only one concentration of SO and OSO supplementation in chronic treatment experiments?

- Was the fixed liver tissue embedded in the parafilm????? Maybe it's "paraffin"? Check for grammar and spelling errors and carelessness!

- Results are written garbled. For example, in line 184-191, while discussing the effects of OSO the authors switch to the activity of SO1% and then go back and repeat that OSO has positive effects on the development of the embryo. No information on SO2%. Perhaps the authors can describe the results more clearly. Similar confusions are also present in the subsequent paragraphs of the results.

- "The fluorescence intensity (FI) from AO and DHE staining indicated that severe ROS production and apoptosis, respectively, occurred in the embryos" Perhaps it is "apoptosis and ROS production, respectively".

The English Language is fine. Only some grammar and spelling errors are present.

Author Response

(The authors gave the same response as above.)

Reviewer 3 Report

Dear,

Thank you for the opportunity to review this paper.

The manuscript entitled “Long-term supplementation of ozonated sunflower oil improves dyslipidemia and hepatic inflammation in hyperlipidemic zebrafish: Suppression of oxidative stress and inflammation against carboxymethyllysine toxicity” had as aim to investigate the lipid-lowering effects of ozonated sunflower oil in high-cholesterol diet-induced hyperlipidemic zebrafish, namely in the improvement of hepatic inflammation, suppression of oxidative stress and inflammation against carboxymethyllysine toxicity.

In my opinion, this study needs improvement and clarification, as sent to authors, particularly in the methodology.

Author Response

(The authors gave the same response as above.)

Round 2

Reviewer 1 Report

The auhtors have significantly improved the manuscript by replying satisfactorily and including new data therefore improving the sientific merit of the work.

Reviewer 2 Report

The authors have answered all my requests and the article is ready for publication.